# Evidence for social parasitism of early insect societies by Cretaceous rove beetles

Shûhei Yamamoto[1,*], Munetoshi Maruyama[2] & Joseph Parker[3,4,*]

The evolution of eusociality in ants and termites propelled both insect groups to their modern ecological dominance. Yet, eusociality also fostered the evolution of social parasitism—an adverse symbiosis, in which the superorganismal colonies formed by these insects are infiltrated by a profusion of invertebrate species that target nest resources. Predominant among these are the aleocharine rove beetles (Staphylinidae), a vast and ecologically diverse subfamily with numerous morphologically and behaviourally specialized socially parasitic lineages. Here, we report a fossil aleocharine, *Mesosymbion compactus* gen. et sp. nov., in Burmese amber (~99 million years old), displaying specialized anatomy that is a hallmark of social parasites. *Mesosymbion* coexisted in the Burmese palaeofauna with stem-group ants and termites that provide the earliest indications of eusociality in both insect groups. We infer that the advent of eusociality led automatically and unavoidably to selection for social parasitism. The antiquity and adaptive flexibility of aleocharines made them among the first organisms to engage in this type of symbiosis.

[1] Entomological Laboratory, Graduate School of Bioresource and Bioenvironmental Sciences, Kyushu University, Hakozaki 6-10-1, Higashi-ku, Fukuoka 812-8581, Japan. [2] The Kyushu University Museum, Hakozaki 6-10-1, Higashi-ku, Fukuoka 812-8581, Japan. [3] Department of Genetics and Development, Columbia University, 701 West 168th Street, New York, New York 10032, USA. [4] Division of Invertebrate Zoology, American Museum of Natural History, New York, New York 10024, USA. * These authors contributed equally to this work. Correspondence and requests for materials should be addressed to J.P. (email: jp2488@columbia.edu).

With over 61,300 species[1], rove beetles (Staphylinidae) are currently recognized as the most speciose animal family. Among the 32 extant subfamilies, Aleocharinae is the largest and accounts for approximately one quarter of rove beetle diversity[2]. The subfamily represents one of Metazoa's great radiations, a feat achieved in part by dramatic habitat, microecological and behavioural specialization in numerous lineages[3]. Most aleocharines are minute (typically 2–6 mm) predaceous inhabitants of leaf litter and soil microhabitats, but the group has colonized almost every terrestrial niche, including birds' nests, caves[4], intertidal zones and exposed coral reefs[5], fungi, and the surfaces of vascular plants[6]. Various groups have undergone shifts to mycophagy, saprophagy and palynophagy[3], as well as to ectoparasitoidism[7], but the adaptive flexibility of aleocharines is most striking in the numerous lineages that have transitioned to life inside social insect colonies. Such taxa live as socially parasitic myrmecophiles or termitophiles—specialized guests that exploit nest resources, prey on the brood, and in some cases achieve social integration where they are accepted as nestmates[8–12]. Evolution of this way of life has arisen convergently across the subfamily[12], leading to major changes in anatomy[8,9,12], glandular chemistry[13] and behaviour[14,15] that adjust the beetles to an obligate, symbiotic existence.

Aleocharines have arguably been the most successful arthropod group at capitalizing on the ecological dominance of eusocial insects[4,12], with previous authors reasoning that some socially parasitic relationships within the subfamily are ancient, extending back to near the origins of termites and ants in the Mesozoic[8,10,16]. This antiquity has been inferred indirectly: social parasitism occurs in some primitive aleocharine lineages of presumed Mesozoic origin[10], and the conserved associations of certain aleocharine groups with the same ant or termite subfamily across broad, often pantropical zoogeographic ranges have been interpreted as arising from Pangaean or Gondwanan vicariance[8–10,16–19]. Yet, gauging whether this type of symbiosis was a bona fide feature of early ant and termite ecology has proven challenging for several reasons. First, although termites are believed to have evolved in the Late Jurassic[20,21], and ants somewhat later in the Early Cretaceous[22–25], their frequency in fossil deposits implies that both taxa remained rare for much of their early evolution. Each group comprises <1% of all insect fossils in any given Cretaceous locality[20,22,24], and only in the Cenozoic do both groups increase dramatically towards their modern abundances. The probability of recovering fossil social parasites of these rare Mesozoic ants and termites is extremely small; myrmecophiles and termitophiles typically exist at densities orders of magnitude lower than their hosts[10], and consequently, no socially parasitic aleocharines have been reported before the Miocene[16,26]. Moreover, the earliest-known social parasite belonging to any arthropod group—the myrmecophile *Protoclaviger trichodens* Parker and Grimaldi, a pselaphine rove beetle—is known from the Early Eocene (~52 Mya)[27]. Second, the precondition for the evolution of social parasitism is the existence of resource-rich nests that invite exploitation. However, undisputed evidence that Mesozoic termites and ants were definitively eusocial has until very recently been lacking[21,28–30], creating uncertainty as to whether either taxon formed colonies. Finally, ambiguity lies in whether aleocharines are genuinely an ancient enough group to have evolved social parasitism at this early time. While the subfamily is relatively common in Cenozoic deposits, the single Mesozoic (mid-Cretaceous) species thus far reported belongs to the entirely free-living, basal lineage of Aleocharinae[31], providing no insight into when the socially parasitic groups in the remainder of the subfamily might have arisen.

Here, we address these uncertainties directly with a new and unusual aleocharine fossil in mid-Cretaceous Burmese amber, dated to 98.8 million years old (earliest Cenomanian)[32]. The specimen represents a new genus and species, and is remarkable in exhibiting defensive modifications that are hallmarks of a socially parasitic lifestyle, and which have evolved convergently multiple times in modern aleocharines as well as in myrmecophiles and termitophiles scattered across other staphylinid subfamilies. Recent studies of stem-group ants and termites in Burmese amber report clear evidence of advanced social organization in both insect groups by the mid-Cretaceous[21,25]. The new fossil taxon indicates that early colonies formed by these insects were targeted by specialized social parasites, extending the age of this kind of symbiosis back by ∼50 million years, close to the inferred advent of ant and termite eusociality.

## Results

### Systematic palaeontology.

Order Coleoptera Linnaeus, 1758
Superfamily Staphylinoidea Latreille, 1802
Family Staphylinidae Latreille, 1802
Subfamily Aleocharinae Fleming, 1821
Tribe Mesoporini Cameron, 1959
**Mesosymbion compactus** gen. et sp. nov.

**Diagnosis of new genus and species**. Mesoporine aleocharines distinguished from all other genera of Mesoporini by the possession of short, thick, clavate antennal flagellae, with extremely transverse antennomeres 4–10 that appear to telescope, with the base of one antennomere secluded by the apex of the previous one so that the pedicels are concealed (Fig. 1f,g, Supplementary Fig. 1c,d, Supplementary Video 3); head triangular, opisthognathous with mandibles pointing posteriorly, completely hidden under the pronotum and not visible in dorsal view (Fig. 1a–d, Supplementary Fig. 1a, Supplementary Video 4); antennae inserted under shelf-like margins of the frons (Fig. 1d, Supplementary Video 4); mandibles slender, falciform, lacking apical inner teeth, their bases contiguous so that they appear like crossed shears when closed (Fig. 1d, Supplementary Fig. 1a, Supplementary Video 4); maxillary palpi with small palpomere 4, only 1/3 as long as palpomere 3 (Fig. 1e); Mesosternal intercoxal process sharply pointed, with its apex lying slightly over that of metasternal intercoxal process (Supplementary Fig. 1e). See Supplementary Note 1 for a full description of the new genus and species. The phylogenetic position of the new genus and species is shown in Fig. 2.

**Age**. Upper Cretaceous (earliest Cenomanian; 98.8 Mya[32]). See Supplementary Note 1 for Geographic and Geological Context.

**Holotype material**. Sex unknown (putative male). Data label: 'AMBER: MYANMAR (BURMA), Upper Cretaceous, Kachin: Noije Bum mines, near Tanai Village (105 km NW Myitkyina), AMNH Bu-SY5'. Specimen in the American Museum of Natural History.

**Etymology**. The generic name is a combination of 'Mesozoic' and the Greek 'συν' (Syn) and 'βος' (Bios) meaning 'living together' in reference to the probable symbiotic ecology of the new taxon during the Mesozoic era. The gender is masculine. The specific name is Latin for 'compacted' on account of the compact, limuloid body plan and antennae.

**Systematic position**. Staphylinidae is informally divided into four subfamily groups[3,33]. *Mesosymbion* belongs in the

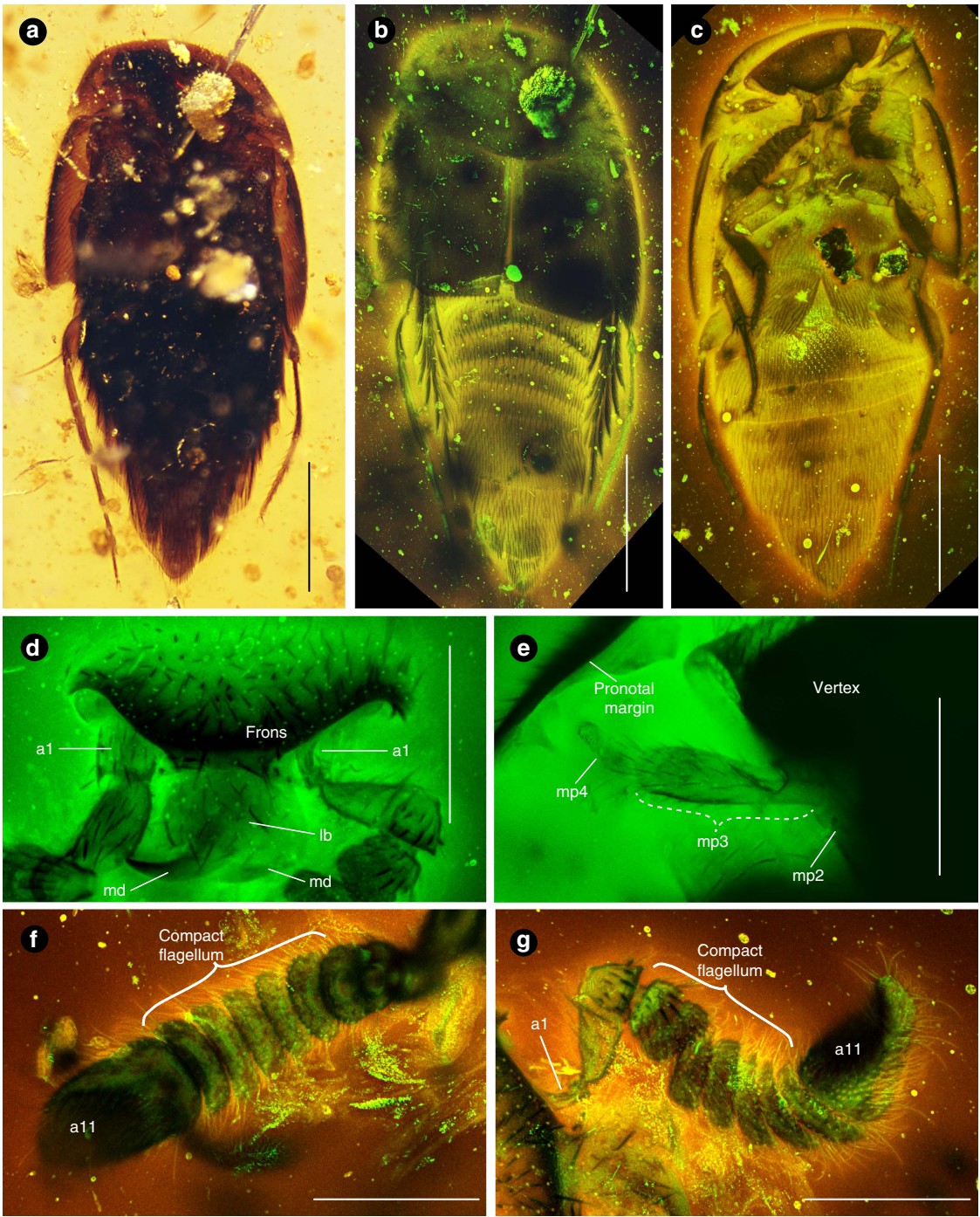

**Figure 1 | *Mesosymbion compactus* genus et species nova.** (**a**–**c**) Habitus images of holotype AMNH Bu-SY5. Dorsal habitus, light microscopy (**a**), dorsal habitus, confocal reconstruction (**b**), ventral habitus, confocal (**c**). (**d**) Confocal image of frons showing falciform mandibles (md), labium (lb) and shielded antennal insertions (a1: antennomere 1). (**e**) Right maxillary palpus with palpomeres (mp2–mp4) indicated. (**f**,**g**) Left and right antennomeres, respectively, showing compaction of antennomeres 4–10 (a11: antennomere 11). Scale bars in **a**-**c**, 250 μm; scale bars in **d**-**g**, 75 μm.

tachyporine group due to its fully limuloid body shape (Figs 1a–c and 3a; discussed below) and tapered abdomen with six visible sternites (excluding genital segments; Fig. 1b, Supplementary Videos 1 and 2). The tachyporine group is composed of six subfamilies: Aleocharinae, Habrocerinae, Olisthaerinae, Phloeocharinae, Tachyporinae and Trichophyinae[34]. *Mesosymbion* is placed within Aleocharinae on the basis of its clavate antennae (Fig. 1f,g, Supplementary Video 3), strongly sinuate posterolateral elytral margins[35] (Figs 1b and 3a, Supplementary Video 1), plus an overall habitus consistent with this subfamily. Aleocharines usually possess antennal insertions anterior to the eyes on the head vertex, but derived modifications to the head of *Mesosymbion* that we believe perform a defensive function mean that the antennal insertions are shielded by the overhanging frons (Fig. 1d, Supplementary Video 4), and are hence uncharacteristic of the subfamily[3,36,37].

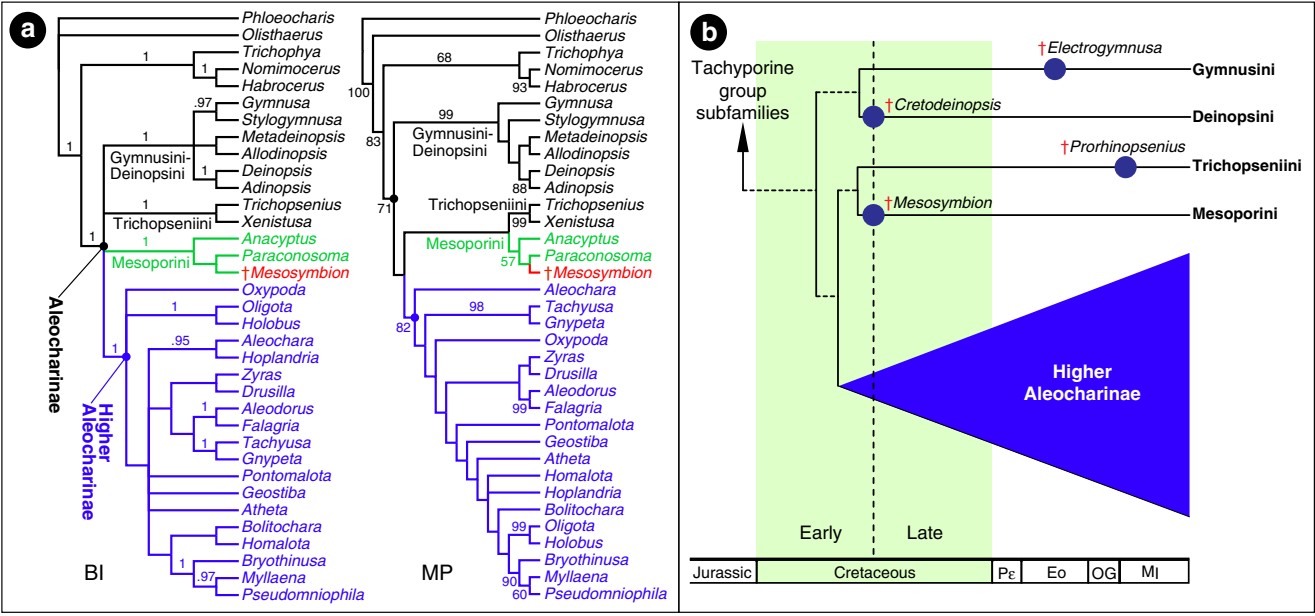

**Figure 2 | Phylogenetic position of *Mesosymbion* and implications for the timeframe of aleocharine evolution.** (**a**) Consensus trees from Bayesian inference (BI) and maximum parsimony (MP) analysis of *Mesosymbion* and representative aleocharines and tachyporine group taxa. Posterior probabilities above 0.9 are shown on branches in the BI tree; bootstrap percentages >50 from 10,000 replicates are shown on the MP tree (strict consensus of two trees: length 542 steps, CI = 0.45, RI = 0.65). (**b**) Proposed scenario of early cladogenetic events in Aleocharinae, employing the topology from Ashe[34] and our MP analysis in **a**, scaled by the earliest-known fossils of Deinopsini and Mesoporini (blue circles indicate oldest fossils known for each basal lineage outside of the higher Aleocharinae).

Aleocharinae is split into a so-called 'higher Aleocharinae' diagnosed by the unique presence of a dorsal defensive gland between abdominal tergites VI and VII in adults[34,38], and a basal glandless-grade of ~140 species split into four tribes: Gymnusini, Deinopsini, Trichopseniini and Mesoporini. Phylogenetic work on aleocharine relationships by Ashe[34] proposed a clade comprising Gymnusini + Deinopsini as the earliest diverging lineage of the subfamily, followed by Trichopseniini + Mesoporini as the immediate sister to the higher Aleocharinae (Fig. 2a,b). These latter two tribes today are composed exclusively (Trichopseniini[8,16,39]) or in part (Mesoporini[8,40–42]) of species that are symbionts of termite colonies, and their basal position makes them strong contenders for having evolved social parasitism in the Mesozoic. We place *Mesosymbion* into Mesoporini based on its small, limuloid body with heavily reticulated sculpturation (Supplementary Fig. 1g,g′), 5-5-5 tarsal formula (Supplementary Fig. 1f), mesoventrite expanded and shield-like (*sensu* ref. 34) and presence of a hindcoxal lamella (Supplementary Fig. 1g,g′, Supplementary Video 5). A similar set of character states occur in Trichopseniini (some limuloid genera of which bear a resemblance to *Mesosymbion* that extends to its defensive head morphology), as well as the higher aleocharine tribe Hypocyphtini. However, in Trichopseniini the lamella is united with the metaventrite to form a metasternal plate[16,39], and abdominal segment IX is strongly subdivided. Hypocyphtini differ to *Mesosymbion* in their possession of 10-segmented antennae and 4-segmented tarsi. Consistent with our *a priori* judgement that *Mesosymbion* is a mesoporine, both Bayesian and parsimony analysis using Ashe's character matrix[34] augmented with characters diagnostic for Mesoporini places the new taxon firmly inside the Aleocharinae, and within Mesoporini (Fig. 2a, Supplementary Fig. 2; for further discussion of the new taxon's position within Aleocharinae and Mesoporini, see 'Systematic Position of *Mesosymbion*' in

Supplementary Note 1). The age, phylogenetic position and morphology of *Mesosymbion* collectively provide evidence that social insect colonies were targeted by socially parasitic aleocharines in the Mesozoic.

**Mesosymbion and Cretaceous evidence of social parasitism.** The earliest-known ants, as well as the earliest-known morphologically specialized termite castes occur in Burmese amber, with this deposit providing the clearest evidence that both insect groups had evolved an advanced state of eusociality by the mid-Cretaceous[21,25]. Remarkably, *Mesosymbion*, recovered from this same amber deposit, bears a specific set of morphological hallmarks that indicate it was an obligate social parasite of the early colonies formed by either ants or termites in the Burmese palaeofauna. The body shape, thoracic morphology and cephalic modifications (in particular the head shape, orientation and antennal form) signify an ecomorphological syndrome that has evolved convergently in multiple clades of myrmecophiles and termitophiles in Aleocharinae.

First, *Mesosymbion* possesses a defensive 'limuloid' (horseshoe crab- or teardrop-shaped) body plan[10,12], where the pronotum is expanded anteriorly and laterally to form a protective hood, underneath which the head and appendages can be retracted (Figs 1a–c and 3a; Supplementary Video 2; the head and all appendages except the hind legs are shielded from above in *Mesosymbion*). This body shape has arisen independently numerous times in myrmecophiles and termitophiles belonging to Aleocharinae, as well in a handful of socially parasitic taxa from other staphylinid subfamilies[8–10,43,44] (Fig. 3c–g). Species with this morphology are typically not socially integrated inside colonies, and are treated aggressively by their hosts, at least during part of their adult stage[10]. The exaggerated 'full' version of this morphology, where the pronotum completely covers the head in dorsal view, is a characteristic of such socially parasitic taxa[8]. In exceptional limuloid genera, a further modification has occurred where the head orientation has rotated by almost 180°, from ancestrally

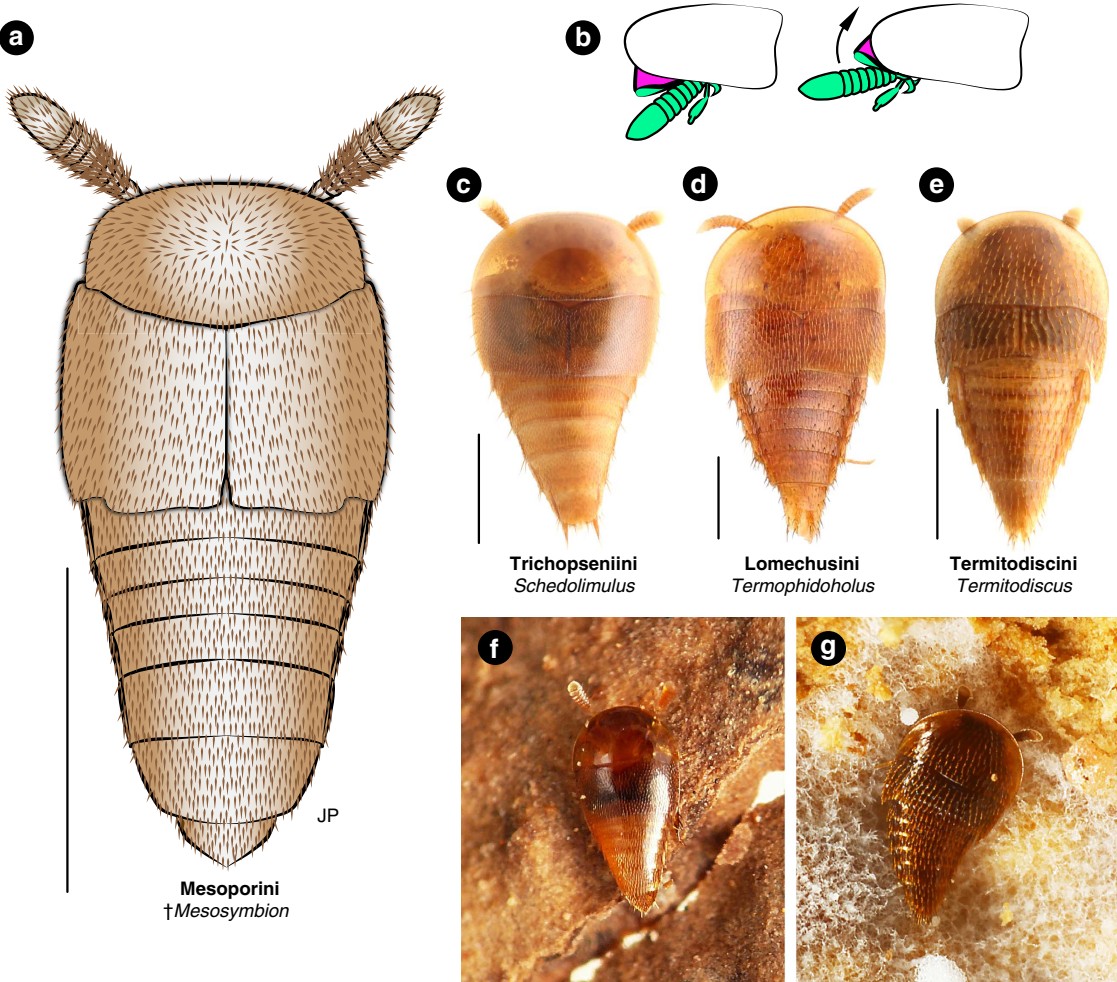

**Figure 3 | Convergent evolution of the limuloid body plan in Aleocharinae.** (**a**) Model reconstruction of *Mesosymbion compactus*, dorsal habitus with antennae extended. (**b**) Hypothetical articulation of the head of *Mesosymbion* (green), from fully retracted under the pronotum to raised, partially extending beyond the pronotum and exposing the occiput (magenta). (**c–e**) Representative convergent limuloid aleocharines (all termitophilous) from phylogenetically distant tribes: *Schedolimulus* sp. (Trichopseniini) (**c**); *Termophidoholus formosanus* (Lomechusini: Termitozyrina) (**d**); *Termitodiscus* sp. (Termitodiscini) (**e**); images courtesy of Taisuke Kanao. (**d**) was previously published in ref. 65. (**f**,**g**) Living termitophiles inside termite nests: *Schedolimulus komatsui* on carton wall of *Schedorhinotermes* sp. nest (**f**); *Termitodiscus* sp. in fungus garden of *Odontotermes* sp. host (**g**); images courtesy of Takashi Komatsu. (**f**) was previously published in ref. 66. Scale bars in **a**,**c**,**d**,**e**, 500 μm.

prognathous/weakly hypognathous (pointing forward or slightly downward; Supplementary Fig. 3a,b), to opisthognathous (pointing backwards), where the mandibles are directed caudally and the occiput (back of the head) forms a continuous shield with the pronotum hood[8,9,45] (Supplementary Fig. 3c–e). Remarkably, not only is the head of *Mesosymbion* able to retract fully under the limuloid pronotum (Figs 1a–c), but it is also strongly opisthognathous (Fig. 1c, Supplementary Fig. 1a, Supplementary Video 4), and we think it may be able to extend somewhat more anteriorly and still be shielded by the occiput (Fig. 3b).

To our knowledge, this extreme pronotal and cephalic reconfiguration is only seen in specialized socially parasitic genera in Aleocharinae such as the termitophile *Athexenia*[45], as well as in the strongly limuloid army ant-associated tachyporine, *Vatesus*[9]. *Mesosymbion* thus displays the earliest-known example of the fully limuloid body plan that is characteristic of obligate social parasites, and moreover an advanced version of it. We posit that initial Mesozoic evolution of this morphology may have been facilitated by the prior evolution in free-living ancestors of a quasi-limuloid body shape, where the pronotum is expanded but only partially covers the head and appendages (Supplementary Fig. 3a). This condition, seen in basal, free-living aleocharines of the tribes Gymnusini and Deinopsini, may represent the ancestral state in the subfamily—one that was coopted for the convergent evolution of the strongly limuloid shape of *Mesosymbion*, as well as the definitively socially parasitic mesoporine sister tribe, Trichopseniini (Fig. 3c,f). During subsequent evolution of the subfamily, quasi-limuloid morphology appears to have been primitively lost in some major clades of higher Aleocharinae, such as the Athetini-Lomechusini assemblage of tribes[46] (Supplementary Fig. 3b), but fully limuloid morphology has re-evolved in multiple instances where taxa have transitioned to social parasitism (such as in the Lomechusini subtribe, Termitozyrina; Fig. 3d). Correlated with the evolution of fully limuloid morphology is a shortening and broadening of the body into a compact teardrop shape, with a reduced abdomen length and a widening of the body across the elytra. In the basal, quasi-limuloid, free-living aleocharine *Gymnusa*, body width across the elytra is half ($\times 0.52$) the extended abdomen length (Supplementary Fig. 3a). In *Mesosymbion* and the termitophiles in Fig. 3c–e, body width and abdomen length are approximately equal (ratios of 0.93, 1.2, 1.0 and 0.94, respectively).

An equally prominent defensive adaptation is seen in the antennal morphology of *Mesosymbion*. The antennae of free-

living aleocharines are elongate, thin and filiform or weakly clavate in shape (for example, Supplementary Fig. 3a,b). Convergently in socially parasitic groups, however, the antenna has undergone reinforcement, presumably to safeguard against losing the appendage to host aggression[10]. In such cases, the segments become strongly transverse, transforming the antenna into a thickened, robust club-like shape, where the antennomeres are compacted into each other[12] (Fig. 3c–g). By reducing or concealing the weaker, connecting antennomere pedicels, the segments appear to telescope, with the base of each segment nested inside the segment preceding it (Fig. 1f,g; Supplementary Fig. 1c,d and Supplementary Video 3 show confocal sections taken on the surface of the right antenna compared with a sagittal section through the antenna, emphasizing the nesting antennomeres with concealed pedicels). This antennal form is a hallmark of many independently-evolved aleocharine symbionts of both ants and termites, as well as some social parasites occurring in other staphylinid subfamilies, including Tachyporinae (Vatesus; Supplementary Fig. 3e)[47], Pselaphinae (Attapsenius)[12] and Scydmaeninae (Plaumaniolla)[48]. In many limuloid social parasites that have such robust, compact antennae, including members of Pseudoperinthini, Pygostenini, Termitodiscini, Termitohospitini, Termitonannini, Trichopseniini and others[8], the antennae are also shortened so that they can be mostly concealed under the hood-like pronotum (Fig. 3c–g). Such antennae, secluded underneath the pronotum, are present in classical form in Mesosymbion (Fig. 1c, Supplementary Fig. 1a, Supplementary Video 4).

Other aspects of the opisthognathous head of Mesosymbion are also derived modifications that attest to its probable socially parasitic biology. The vertex is explanate (laterally expanded) at the margin to shield the antennal insertion points (Fig. 1d, Supplementary Video 4), and the mandibles are falciform (thin hooks) without internal teeth, and have their bases extremely close together (Fig. 1d, Supplementary Video 4). This scissor-like morphology suggests these are not the typical predatory, raptorial mandibles with a strong compressing force seen in many free-living predatory staphylinids[49]. With the head fully concealed under the pronotum and the mandibles directed caudally, it is unlikely that Mesosymbion targeted moving prey, and the mandibular form is indicative of rove beetles that feed by chewing soft, immobile objects such as Dipteran larvae[49] and fungi[50], or in the case of social parasites, the eggs, larvae and pupae within brood galleries. Indeed, comparing Mesosymbion with modern social parasites with detailed ecological data, the combination of a fully limuloid body, compact antennae with concealed/absent pedicels, an opisthognathous, triangular head with eyes mounted at the apices of the triangle (Supplementary Fig. 1a,b) and falciform mandibles pointing caudally is perhaps most closely convergent with Vatesus[9] (Supplementary Fig. 3c–e), a specialized symbiont of Neotropical army ants[51] that feeds on the colony brood[52].

Together, the suite of characters presented by Mesosymbion define an ecomorphology that has arisen numerous times in Aleocharinae, and suggests a non-integrated social parasite that was probably treated aggressively by its hosts, potentially targeting colonies as a brood predator. Obvious morphological specializations that in aleocharines indicate social integration and acceptance inside host colonies are absent from Mesosymbion. The taxon lacks the physogastric (swollen) body form of some aleocharine termitophiles[8,10,17], or a myrmecoid (ant-like) body shape that is seen in army ant-associated groups[9,10,12]. Evidence of glandular complexes or trichomes associated with production of appeasement compounds is also missing[12,15]—this highly intimate form of social parasitism thus far dates to the Early Eocene, manifested in the trichome-bearing pselaphine

Protoclaviger[27]. Before this study, the earliest-known aleocharines with comparable morphology to Mesosymbion are in Miocene Dominican and Mexican ambers[16,26].

With only 19 species, Mesoporini is one of the smallest and rarest groups of aleocharines. Little detailed information exists on the biology and ecological habits of most species, but social parasitism in the form of termitophily has nevertheless been observed in half of the extant mesoporine genera: Mesoporus[40], Dictyon[41], Anacyptus[8] and Kistnerium[42]. Hence, a socially parasitic biology of Mesosymbion is supported by both its morphology and a recurring trend of termite associations in the tribe to which it belongs. Notably, despite its antiquity, Mesosymbion has an anatomy more overtly specialized for social parasitism than other inquilinous mesoporines, which lack the fossil taxon's extreme head and antennal modifications. Mesosymbion may thus have been correspondingly more closely associated with its hosts than are Recent termitophilous Mesoporini. Given that the putative sister group of Mesoporini is Trichopseniini, an exclusively termitophilous tribe, we posit that either the Mesoporini-Trichopseniini clade is predisposed to this way of life and has repeatedly evolved it, or that termitophily is ancestral and has been lost in some mesoporine genera that are believed to be free-living, such as Paraconosoma. On these grounds, we suspect that if Mesosymbion was indeed a social parasite as its morphology implies, it was probably a termitophile. Termites are a more ancient eusocial group than ants[20,21], raising the likelihood that by the mid-Cretaceous their nests would have succumbed to social parasites.

## Discussion

Before the discovery of Mesosymbion, the single known Mesozoic aleocharine was Cretodeinopsis aenigmatica Cai & Huang, also recovered from Burmese amber[31] (see Supplementary Table 1 for an inventory of described fossil Aleocharinae). Cretodeinopsis belongs to the tribe Deinopsini, part of the earliest diverging clade of Aleocharinae. Although Cretodeinopsis extends the age of the subfamily to the mid-Cretaceous, its basal position within the subfamily left open the question of when other early cladogenetic events within Aleocharinae occurred. Assuming the topology of basal aleocharine relationships of Ashe[34] is correct, Mesosymbion reveals that all three major clades—Gymnusini + Deinopsini, Trichopseniini + Mesoporini and the higher Aleocharinae (either stem- or crown-group)—date to at least the mid-Cretaceous (Fig. 2b). We consequently infer that the divergences leading to these three clades happened before this time, during the Early Cretaceous at the latest (Fig. 2b). Such a timescale fits with molecular dating analysis of a large sampling of aleocharine tribes that has yielded ages for Trichopseniini and the higher Aleocharinae of ~108 Mya and ~110 Mya, respectively[53]. A significant fraction of Mesoporini genera[8,40–42] and all modern members of Trichopseniini[8,16,39] are known to associate with termites. In addition, myrmecophily and termitophily have evolved dozens (perhaps hundreds) of times independently across the higher Aleocharinae[8–12]. Hence, the observed or inferred presence of all of these groups in the mid-Cretaceous makes it possible that Mesosymbion was not alone in targeting colonies, and that multiple aleocharine taxa were social parasites at this time. In addition, crown-group Pselaphinae—another rove beetle subfamily equally predisposed to ant and termite exploitation[12]—have been described from Burmese amber[54]. It may be that by the mid-Cretaceous, social insects already possessed a 'bestiary' of social parasites[11].

Why have aleocharines, as opposed to almost any other insect group, been so successful at invading colonies of social insects? We have previously argued that a predatory diet, physically or

chemically defensive morphology and small body size are preadaptive traits that have synergized to make the subfamily especially prone to evolving social parasitism[12]. *Mesosymbion* reveals that this adaptive versatility extends deep into the Mesozoic, when eusocial colonies presented novel niches for occupation that few other taxa were equivalently predisposed to fill. The notion of Mesozoic social parasitism by aleocharines implies that ant and termite societies were subject to exploitation during most of their evolution, including a long period when both social insect groups are inferred to have been rare and ecologically insignificant[20,22,24]. We propose that, despite their apparent scarcity, evolution of the resource-rich colonies of both ants and termites immediately engendered selection for social parasitism; it is an unavoidable counterpart of eusociality. It is possible—in fact probable—that this type of symbiosis dates to an even earlier time in the evolution of both eusocial groups.

## Methods

**Specimen imaging and description.** During our survey of staphylinids in Burmese amber, we discovered an unusual specimen that was a putative member of the subfamily Aleocharinae. The holotype of the new genus and species is a complete specimen contained in a small triangular fragment that was cut and polished by the first author (S.Y.) and Y. Takahashi (University of Tsukuba, Tsukuba, Japan), revealing dorsal and ventral views of the body. The inclusion is deposited in the American Museum of Natural History (AMNH: D. Grimaldi, curator), New York, USA, with specimen number AMNH-SY5. The beetle is well preserved (Fig. 1a), but challenging to observe due to its dark pigmentation. We employed multi-channel laser-scanning confocal microscopy, combining this technique with multiple image montage projection that is typically used for light microscopy images. Montage projections of narrow-plane confocal Z-stacks revealed the anatomy of the new specimen with high resolution. A Leica SP5 confocal microscope with 488, 543 and 647 nm lasers and HyD detectors was used to create image stacks, with the combination of lasers varying depending on the structure being imaged. Zerene Stacker was subsequently used to produce montage images, and image stacks were also exported from Leica LAS AF software as movies, which facilitated detailed observation of the specimen and enabled us to formally describe it. Raw confocal micrographs are available on request from the corresponding author (J.P.). For morphological description of the new taxon, we used the terminology of Newton *et al.*[36] and Thayer[3].

**Phylogenetic analysis.** We scored *Mesosymbion* for 160 characters used by Ashe[34] in a study of basal relationships in Aleocharinae, and integrated these characters with the matrix used in that study, which includes representatives of Mesoporini, Gymnusini, Deinopsini, Trichopseniini, numerous genera of higher Aleocharinae and non-aleocharine outgroups. We excluded the taxa belonging to the subfamily Tachyporinae, the most basally-nested lineage in Ashe[34] which helped to stabilize parts of the Aleocharinae clade; including Tachyporinae did not affect the placement of *Mesosymbion* within Mesoporini. The matrix was constructed in Mesquite v. 3.10 (ref. 55) and is presented below in Supplementary Table 2, as well as in the MrBayes nexus file (Supplementary Data 1). We revised character 120 to accommodate the antennal form of *Mesosymbion* and added four new characters to the matrix that are relevant to the diagnosis of Mesoporini (characters are expressed as homology statements following Sereno[56], and run from character 0—character 163, following Ashe[34]):

*Character 119*: antenna, overall shape: (1) apical articles not enlarged to form an apical 'club'; (2) subapical articles moderately enlarged to form a loose 'club'; (3) subapical articles strongly enlarged to form a distinct 'club'; (4) overall antennal shape strongly clavate.

*Character 160*: body, length: (1) medium to long (≥1.35 mm; 'typical' aleocharine size range); (2) distinctly short (<1.35 mm).

*Character 161*: abdomen, sternites, reticulation: (1) absent; (2) present (Fig. 1c, Supplementary Fig. 1g,g').

*Character 162*: antennal insertions, dorsal view: (1) more or less visible from above but partially concealed by frontal shelf; (2) fully exposed from above, lacking frontal shelf (Supplementary Fig. 3b); (3) more or less visible from above but partially concealed by frontal shelf, but developed frontal shelf also conceals half of antennomere I (Fig. 1d).

*Character 163*: aedeagus, male: (1) paramerite fused or tightly attached to median lobe; (2) paramerite loosely attached and can be easily removed from median lobe; (3) aedeagus highly modified[57].

All characters were non-additive, unordered and equally weighted. Bayesian analysis was carried out using MrBayes 3.2.3 (ref. 58) accessed via the Cipres Science Gateway[59]. The Mkv + G model[60] was specified, and two MCMC runs of four chains were run for two million generations. Convergence was judged to have occurred when the standard deviation of split frequencies dropped below 0.005,

and by ESS values higher than 200 in Tracer[61], indicating adequate estimation of the posterior. The first 25% of trees were discarded as burn-in. Parsimony analysis was performer using TNT[62] using New Technology search, and branch support values were estimated using 10,000 bootstrap replicates[63]. Mapping character states onto the phylogeny was performed with WinClada[64].

**Nomenclatural acts.** This published work and the nomenclatural acts it contains have been registered in ZooBank, the proposed online registration system for the International Code of Zoological Nomenclature. The ZooBank LSIDs (Life Science Identifiers) can be resolved and the associated information viewed through any standard web browser by appending the LSID to the prefix 'http://zoobank.org/'. The LSIDs for this publication are to be found at: urn:lsid:zoobank.org:pub: 63EA52A4-765A-4349-A74A-A1032909BA39

**Data availability.** All data generated or analysed during this study are included in this published article (and its Supplementary Information files). The holotype specimen of *Mesosymbion compactus*, around which this study is based, is deposited in the American Museum of Natural History, New York (accession number AMNH Bu-SY5).

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

## Acknowledgements

We thank Yui Takahashi (University of Tsukuba) for preparing the *Mesosymbion* specimen and David Grimaldi (AMNH) for the providing the accession number of the holotype for deposition. Alfred Newton and Margaret Thayer (Field Museum of Natural History), Taro Eldredge (University of Kansas), Christoph von Beeren (Technische Universität Darmstadt) and Toshiya Hirowatari (Kyushu University) provided important feedback on the manuscript. We are grateful to Takashi Komatsu (Kyushu University), Taisuke Kanao (Kyoto University) and Christoph von Beeren for use of photographs. This study was partially supported by a Grant-in-Aid for JSPS Fellows (14J02669) to S.Y. from the Japan Society for the Promotion of Science, Japan. This is a contribution from the Entomological Laboratory, Faculty of Agriculture, Kyushu University, Fukuoka, Japan (Ser. 7, No. 39). J.P. was funded by a Sir Henry Wellcome Postdoctoral Fellowship and grants from the NIH (RO1 GM113000) and the Ellison Medical Foundation to Gary Struhl, who provided a wonderful environment for this work.

## Author contributions

S.Y. and J.P. conceived and designed the project, with input from M.M. J.P. performed confocal and light microscopic imaging of the amber specimen. S.Y. described the specimen, evaluated its systematic placement. S.Y. and J.P. performed phylogenetic analyses. S.Y. and M.M. photographed extant aleocharine specimens. J.P. wrote the paper and produced figures with input from S.Y. and M.M.

## Additional information

**Competing financial interests**: The authors declare no competing financial interests.

**How to cite this article**: Yamamoto, S. *et al.* Evidence for social parasitism of early insect societies by Cretaceous rove beetles. *Nat. Commun.* **7**, 13658 doi: 10.1038/ncomms13658 (2016).

**Publisher's note**: 

