## [Peer Review File · Nature Communications]

Editorial Note: Two images have been redacted for copyright reasons.

Reviewers' comments:

Reviewer #1 (Remarks to the Author):

I will start my review by saying that I think this is an awesome discovery and a very interesting fossil. However, I think the authors are jumping to conclusions and using problematic methodology for their phylogenetic analyses.

The authors state in the abstract that: "We infer that the advent of eusociality led automatically and unavoidably to selection for social parasitism." The context for the words "automatically" and "unavoidably" are not really explained anywhere in the main text. Moreover, I think this is a statement that has no support based on the evidence presented on this paper.

My biggest concern is this: Does the presence of a character implies the evolution of an associated behavior? The "defensive modifications" could have evolved originally for a different purpose. If this beetle look like other beetles that currently are social parasites, does it mean it is also a social parasite? I think the answer is no, and as much as I want to make the leap of faith here along with the authors, I do not think that it is correct. The problem is that even some extant members of the tribe Mesoporini are not social parasites (according to the discussion provided by the authors in this paper). So, I think it is presumptuous to associate certain morphologies with social parasitism when there are not direct evidence of such parasitism.

I think the discussion on the diagnostic features (or why this fossil must be a social parasite) is too lengthy and honestly not very convincing.

Another problem with this paper is that the phylogenetic analyses were done with parsimony using older software. Even the version of Mesquite used is not current. I think it is unacceptable to simply present parsimony analyses for morphological data - methodology and software for bayesian analyses of morphological data are available for at least the last decade. The strict adherence to parsimony methods is reminiscence of the #parsimonygate and that editorial in Cladistics a few months ago.

Based on the evidence presented and the methodological problems with the phylogenetic analyses, I think this paper belongs in a more specialized journal.

Reviewer #2 (Remarks to the Author):

The manuscript by Yamamoto, Maruyama and Parker describes a newly-discovered aleocharine rove beetle in Burmese amber, dated to 98.8 million years old. Based on the

morphological characters of this specimen, they infer that it was likely a member of the tribe Mesoporinia, and a social parasite. In both respects, it is the earliest known fossil.

The study is of great general interest, and represents a state-of-the-art study of three-dimensional morphology in a difficult-to-access specimen, based on confocal microscopy. The graphics and associated videos make the structures discussed clear, and the conclusions of the authors are well-presented and follow logically from their observations.

The phylogenetic-morphological analysis seems appropriate, but I would have liked to have seen the character matrix included as part of the supplementary material, particularly since it is unclear without accessing further literature which characters were measured and with what certainty.

I have two other small concerns with the manuscript, which should be easily dealt with. One is that technical terms with which the general reader may not be familiar are not always defined when first mentioned (e.g. opisthognathous - which is incidentally spelt "opisthognathus" incorrectly in the legend of Figure S3). The second is that the introductory part of the abstract needs some modification to reflect the field of social parasitism - for example, the statement that social parasites are "solitary arthropod species" is incorrect - they also include many species that are themselves social (in fact some (but a minority of) authors insist that the term social parasite should only be applied to social insects that parasitize other social insect colonies), and also include some non-arthropod species. I would also not describe the aleocharines as "Most exceptional" - if anything, they may be the least exceptional social parasites, because there are so many of them...

Otherwise, the manuscript is well-presented and easy to read, and should be of interest to a wide readership.

Reviewer #3 (Remarks to the Author):

This is a carefully written and illustrated description of an important amber fossil of aleocharine beetle, only the second staphylinid fossil from the Mesozoic and the first such fossil with a limuloid body shape and a suite of other adaptations characteristic of modern symbionts of social insects. The authors argue for the phylogenetic placement of this fossil using comparative biology of staphylinids that this reviewer is not familiar with but trusts given the combined expertise of the authors and the systematists recognized in the acknowledgements.

This is a first-class, very thorough taxonomic description. There is, of course, uncertainty surrounding the inferred phylogenetic placement of the fossil, based upon the morphological dataset of Ashe (2005), his choice of characters, and the shape of the resulting tree. But the lack of bootstrap values above 50 on the tree depicted in Figure 2a indicates there is a lack of statistical support for any node between Mesosymbion and the root of the Aleocharinae. This needs to be addressed directly in the paper, especially as many of their inferences depend directly on this phylogenetic placement. Two examples follow.

(1) Based on Mesosymbion's inferred phylogenetic placement embedded within a clade of termitophiles, the authors suspect that it was also termitophilous.

(2) "Assuming the topology of basal aleocharine relationships of Ashe is correct, Mesosymbion reveals that all three major clades-Gymnusini+Deinopsini, Trichopseniini+Mesoporini, and the higher Aleocharinae (either stem- or crown-group)-date to the mid-Cretaceous (Fig 2b). We consequently infer that the divergences leading to these three clades happened before this time, during the Early Cretaceous at the latest (Fig 2b)."

Are there any dated molecular-based studies of the Aleocharinae that also includes the Trichopseniini and Mesoporini? If so, the paper would benefit greatly from a discussion of the consequences of finding a 99 MYA Mesoporini on our understanding of the evolution of Aleocharinae as a whole.

However, irrespective of its phylogenetic placement, this fossil represents the earliest known putative social insect symbiont of either ants or termites. When social insects evolved and especially after they rose to ecological dominance, they clearly provided a rich resource for those organisms preadapted to take advantage of their nests. Whether or not *Mesosymbion compactum* was a symbiont or it was just preadapted to become one, indicates the social insects were probably parasitized early in their evolution history. As noted by the authors, indications of early evolution of social insect symbionts is also suggested from a suite of dated phylogenies of other groups of social insect parasites, but this is the oldest fossil described to date with morphological features suggesting this lifestyle, which is the significance of this finding.

Reviewer #4 (Remarks to the Author):

This is an interesting study which give compelling evidence for social parasitism by rove beetles already about 100 Ma ago. The preservation of the specimen is remarkable and all important characters are clearly visible and very well documented. The conclusions are convincing and well-balanced. The references are appropriate, even recent paper on termite and ant evolution from 2016 are considered.

I only have a few additional/critical comments:

On page 3 in the first paragraph you write: "However, undisputed evidence that Mesozoic termites and ants were definitively eusocial has been lacking^{21,28-30}, creating uncertainty as to whether either taxon formed colonies."

I do not agree, because meanwhile we have clear evidences for such an early eusociality and indeed, in the next paragraph you write: "Recent studies of stem-group ants and termites in Burmese amber report clear evidence of advanced social organization in both insect groups by the mid-Cretaceous^{21,25}." Here you contradict yourself.

On page 9 in the last paragraph you use the term "ecological niche", but what you mean is that eusocial colonies present a new "resource" for exploitation by rove beetles.

Dear Reviewers,

Thank you for your valuable feedback on our paper. We have addressed each of your comments in the new manuscript. We provide an overview below, as well as a detailed point-by-point rebuttal in the pages that follow (our responses in blue).

Reviewer 2 states the paper "***is of great general interest, and represents a state-of-the-art study***" that is "***well-presented and easy to read, and should be of interest to a wide readership***". Reviewer 2's main points were:

- 1) Include the character matrix as part of the supplementary material. We have now done this.
- 2) Better define the terms "opisthognathous" and "social parasites". We have now clarified our use of these terms.

Reviewer 3 states this is a "***carefully written and illustrated description of an important amber fossil***" that is "***a first-class, very thorough taxonomic description.***" Reviewer 3's main points were:

- 1) Discuss the weak support for *Mesosymbion*'s phylogenetic position. In the revised paper we have redone the analysis with characters appropriate for Mesoporini that Ashe overlooked, which solidifies *Mesosymbion*'s position. We also conducted Bayesian as well as parsimony analyses.
- 2) Discuss possible implications of *Mesosymbion* for understanding/dating Aleocharinae diversification. We now include such a discussion.

Reviewer 4 states our study "***gives compelling evidence for social parasitism by rove beetles already about 100 Ma ago. The preservation of the specimen is remarkable and all important characters are clearly visible and very well documented. The conclusions are convincing and well-balanced***".

Reviewer 4's main points were:

- 1) Clarify our statement that evidence of Mesozoic eusociality has been lacking. In the revised paper we make clear that we meant *until recently*, clear evidence of eusociality had been lacking.
- 2) Replace "niche" with "resource" in our discussion, which we have now done.

Finally, **Reviewer 1** states this is an "***awesome discovery and a very interesting fossil***" but, in contrast to the other reviewers, questions whether the ecology/lifestyle can be inferred from the fossil's morphology. We think this view may stem Reviewer 1's comment that some extant, free-living mesoporines have morphology like *Mesosymbion* (see ***bold, italicized*** text in reviewer 1's point 3 below). This is not the case: *Mesosymbion*'s morphology is a hallmark of specialized socially parasitic species. Free-living aleocharines (including mesoporines) do not have these features. In the new manuscript we have taken every measure to prevent such a misinterpretation by stating in the main text that *Mesosymbion* has specialized anatomy compared to other Mesoporini genera (previously this was mentioned in supplemental information). Reviewer 1 also suggests performing Bayesian analysis. We have now done this.

Reviewer #1:

1) I will start my review by saying that I think this is an awesome discovery and a very interesting fossil. However, I think the authors are jumping to conclusions and using problematic methodology for their phylogenetic analyses. The authors state in the abstract that: "We infer that the advent of eusociality led automatically and unavoidably to selection for social parasitism." The context for the words "automatically" and "unavoidably" are not really explained anywhere in the main text.

Thank you for bringing this up. To clarify, our point is just that because the earliest evidence of social parasitism now coincides with the earliest evidence of eusociality in the fossil record, we think it reasonable that the origin of eusociality automatically set up the selective conditions that engendered the evolution of social parasitism. And now with *Mesosymbion*, already with specialized socially parasitic morphology by ~100 MYA, it is evident that aleocharines were among the first to capitalize on this new resource. We do in fact return to this point explicitly in the paper's final discussion paragraph, where we write: "*We propose that, despite their apparent scarcity, evolution of the resource-rich colonies of both ants and termites immediately engendered selection for social parasitism; it is an unavoidable counterpart of eusociality.*"

2) Moreover, I think this is a statement that has no support based on the evidence presented on this paper. My biggest concern is this: Does the presence of a character implies the evolution of an associated behavior? The "defensive modifications" could have evolved originally for a different purpose. If this beetle look like other beetles that currently are social parasites, does it mean it is also a social parasite? I think the answer is no, and as much as I want to make the leap of faith here along with the authors, I do not think that it is correct.

We respectfully disagree with Reviewer 1's view, which we think may have stemmed from the misunderstanding detailed in point 3 below. Nevertheless, more generally, fossil morphology is arguably a crucial source of evidence for inferences about ancient ecology. Recent examples include the stridulatory file of Jurassic carrion beetles providing evidence of parental care (Cai *et al.* 2014), or distinct termite castes yielding evidence of Cretaceous eusociality (Engel *et al.* 2016). The case of *Mesosymbion* is arguably very strong; those familiar with rove beetles or social parasites in general will know that its suite of morphological traits are a clear hallmark of social parasitism; this morphology is simply not seen in free living species. To emphasise quite how strong this argument is, it is widely accepted among palaeontologists and entomologists alike that fossils with analogous limuloid morphology to *Mesosymbion* in 20-MY old Dominican amber were termitophiles (Grimaldi & Engel 2005; Kistner 1979; 1998; Penney & Jepson 2014; Penney *et al.* 2011; Seevers 1971). *Mesosymbion* now extends the evidence back to 100 MY, coinciding with the earliest evidence of termite and ant eusociality, and *Mesosymbion* itself belongs to the Mesoporini-Trichopseniini clade, which are mostly termitophiles. To our minds, this is as strong and persuasive as any evidence for social parasitism to have come from the fossil record, and its antiquity

means it is arguably an important discovery, as the other reviewers agree. To try to emphasise this point more in the revised paper, we have added the sentence “*Prior to this study, the earliest-known aleocharines with comparable morphology are in Miocene Dominican and Mexican ambers*”.

The literature we have mentioned is as follows, and we show two images from two prominent insect palaeontology books below:

Cai, C.-Y., Thayer, M.K., Engel, M.S., Newton, A.F., Ortega-Blanco, J., Wang, B., Wang, X.-D. & Huang, D.-Y. (2014) Early origin of parental care in Mesozoic carrion beetles. *Proceedings of the National Academy of Sciences* 111, 14170–14174.

Engel, M.S., Barden, P., Riccio, M.L. & Grimaldi, D.A. (2016) Morphologically Specialized Termite Castes and Advanced Sociality in the Early Cretaceous. *Current Biology*, 26, 522–530.

Grimaldi, D.A. & Engel, M.S. (2005) *Evolution of the Insects*. Cambridge University Press.

Kistner, D.H. (1979) *Social and evolutionary significance of social insect symbionts*. In: H. R. Hermann (Ed), *Social Insects*. Academic Press, pp. 339–413.

Kistner, D.H. (1998) New species of termitophilous Trichopseniinae (Coleoptera: Staphylinidae) found with *Mastotermes darwiniensis* in Australia and in Dominican amber. *Sociobiology* 31, 51–64.

Penney, D. & Jepson, J.E. (2014) *Fossil Insects: An introduction to palaeoentomology*. Siri Scientific Press.

Penney, D., Green, D.I. & Marusik, Y.M. (2011) *Fossils in Amber: Remarkable Snapshots of Prehistoric Forest Life*. Siri Scientific Press.

Seevers, C.H. (1971) Fossil Staphylinidae in Tertiary Mexican amber (Coleoptera). *University of California Publications in Entomology* 63, 77–86.

3) The problem is that even some extant members of the tribe Mesoporini are not social parasites (according to the discussion provided by the authors in this paper). So, I think it is presumptuous to associate certain morphologies with social parasitism when there are not direct evidence of such parasitism.

We apologize for the confusion here. To be absolutely clear, one or possibly two genera of Mesoporini are thought to be potentially free-living, but these DO NOT have the fully limuloid/compact antenna+body/modified head and mouthparts of *Mesosymbion*. These extreme morphological modifications are features of highly specialized social parasites only—there are no examples of species with this suite of morphological traits that are free-living. As we mentioned in the suppl. discussion of the previous manuscript, *Mesosymbion* in fact appears more overtly morphologically specialized for social parasitism than all other genera of Mesoporini, with archetypal body modifications of obligate/advanced social parasites belonging to Aleocharinae and other staphylinid/beetle groups. We now mention this point in the main text to avoid any possible ambiguity about this:

“Notably, despite its antiquity, Mesosymbion has an anatomy more overtly specialized for social parasitism than other mesoporines, which lack the fossil taxon’s extreme head and antennal modifications.

4) I think the discussion on the diagnostic features (or why this fossil must be a social parasite) is too lengthy and honestly not very convincing.

We feel strongly that the discussion is a necessary length to explain the morphology in detail, and is consistent with what is the norm in palaeontological papers.

5) Another problem with this paper is that the phylogenetic analyses were done with parsimony using older software. Even the version of Mesquite used is not current. I think it is unacceptable to simply present parsimony analyses for morphological data - methodology and software for bayesian analyses of morphological data are available for at least the last decade. The strict adherence to parsimony methods is reminiscence of the #parsimonygate and that editorial in Cladistics a few months ago.

Thank you for bringing this up. We had simply followed suite with Ashe, which obviously left some room for improvement. In the new paper we include a Bayesian analysis as our primary analysis. Please also see our response to Reviewer 3’s point below.

Reviewer #2:

The manuscript by Yamamoto, Maruyama and Parker describes a newly-discovered aleocharine rove beetle in Burmese amber, dated to 98.8 million years old. Based on the morphological characters of this specimen, they infer that it was likely a member of the tribe Mesoporinia, and a social parasite. In both respects, it is the earliest known fossil. The study is of great general interest, and represents a state-of-the-art study of three-dimensional morphology in a difficult-to-access specimen, based on confocal microscopy. The graphics and associated videos make the structures discussed clear, and the conclusions of the authors are well-presented and follow logically from their observations.

1) The phylogenetic-morphological analysis seems appropriate, but I would have liked to have seen the character matrix included as part of the supplementary material, particularly since it is unclear without accessing further literature which characters were measured and with what certainty.

The matrix is now included in the main part of the supplemental information (as well as in the nexus file, as before).

2) I have two other small concerns with the manuscript, which should be easily dealt with. One is that technical terms with which the general reader may not be familiar are not always defined when first mentioned (e.g. opisthognathous - which is incidentally spelt "opisthognathus" incorrectly in the legend of Figure S3). The second is that the introductory part of the abstract needs some modification to reflect the field of social parasitism - for example, the statement that social parasites are "solitary arthropod species" is incorrect - they also include many species that are themselves social (in fact some (but a minority of) authors insist that the term social parasite should only be applied to social insects that parasitize other social insect colonies), and also include some non-arthropod species. I would also not describe the aleocharines as "Most exceptional" - if anything, they may be the least exceptional social parasites, because there are so many of them. Otherwise, the manuscript is well-presented and easy to read, and should be of interest to a wide readership.

Thank you for bringing these points to our attention. In the new version we have modified the text according to your suggestions. At first mention of "opisthognathous" we write:

"...from ancestrally prognathous/weakly hypognathous (pointing forward or slightly downward; Fig S3a, b), to opisthognathous (pointing backwards), where the mandibles are directed caudally..."

In the abstract we have written:

"...colonies formed by these insects are infiltrated by a profusion of invertebrate species that target nest resources. Predominant among these are the aleocharine rove beetles (Staphylinidae),...."

Reviewer #3:

This is a carefully written and illustrated description of an important amber fossil of aleocharine beetle, only the second staphylinid fossil from the Mesozoic and the first such fossil with a limuloid body shape and a suite of other adaptations characteristic of modern symbionts of social insects. The authors argue for the phylogenetic placement of this fossil using comparative biology of staphylinids that this reviewer is not familiar with but trusts given the combined expertise of the authors and the systematists recognized in the acknowledgements.

1. This is a first-class, very thorough taxonomic description. There is, of course, uncertainty surrounding the inferred phylogenetic placement of the fossil, based upon the morphological dataset of Ashe (2005), his choice of characters, and the shape of the resulting tree. But the lack of bootstrap values above 50 on the tree depicted in Figure 2a indicates there is a lack of statistical support for any node between *Mesosymbion* and the root of the Aleocharinae. This needs to be addressed directly in the paper, especially as many of their inferences depend directly on this phylogenetic placement. Two examples follow.

(1) Based on *Mesosymbion*'s inferred phylogenetic placement embedded within a clade of termitophiles, the authors suspect that it was also termitophilous.

(2) "Assuming the topology of basal aleocharine relationships of Ashe is correct, *Mesosymbion* reveals that all three major clades-Gymnusini+Deinopsini, Trichopseniini+Mesoporini, and the higher Aleocharinae (either stem- or crown-group)-date to the mid-Cretaceous (Fig 2b). We consequently infer that the divergences leading to these three clades happened before this time, during the Early Cretaceous at the latest (Fig 2b)."

A proper evaluation of *Mesosymbion*'s phylogenetic position is indeed desirable, not least because a simple assessment of *Mesosymbion*'s morphology straightforwardly leads to its placement in Mesoporini, a tribe with many termitophiles. Our phylogenetic analysis attempted to strengthen the argument for this tribal placement. However, the analysis itself was relatively simple and as you point out yielded fairly weak support for *Mesosymbion*'s placement. We have now improved the analysis in two ways. First, as you note, Ashe's character matrix was used "as is" without consideration of the types of characters (Ashe was not focused on Mesoporini *per se* and did not include several crucial characters that are diagnostic of this tribe, which we had in fact used in our diagnosis/description to assign *Mesosymbion* to this tribe in the first place). Second, like Ashe, we used parsimony alone, which Reviewer 1 notes is an approach not without problems. In the revised manuscript, we have addressed these issues by including diagnostic characters relevant to Mesoporini, and conducted a reanalysis with both Bayesian and parsimony approaches. The results of both of these analyses lead to a more robust placement of *Mesosymbion* in Mesoporini, consistent with our *a priori* evaluation of the fossil's tribal affinity.

2. Are there any dated molecular-based studies of the Aleocharinae that also includes

the Trichopseniini and Mesoporini? If so, the paper would benefit greatly from a discussion of the consequences of finding a 99 MYA Mesoporini on our understanding of the evolution of Aleocharinae as a whole.

Currently there is no dated Aleocharinae phylogeny, but in the paper we now discuss an unpublished piece of work by two of us (Maruyama and Parker, on convergent evolution of army ant-mimicking aleocharines) where we have performed molecular dating of a large sampling of aleocharine tribes. This work has inferred an age estimate of the higher Aleocharinae of ~110MY, consistent with the tree in Figure 2b, and our paper's text that you quoted in your example (2), above.

However, irrespective of its phylogenetic placement, this fossil represents the earliest known putative social insect symbiont of either ants or termites. When social insects evolved and especially after they rose to ecological dominance, they clearly provided a rich resource for those organisms preadapted to take advantage of their nests. Whether or not *Mesosymbion compactum* was a symbiont or it was just preadapted to become one, indicates the social insects were probably parasitized early in their evolution history. As noted by the authors, indications of early evolution of social insect symbionts is also suggested from a suite of dated phylogenies of other groups of social insect parasites, but this is the oldest fossil described to date with morphological features suggesting this lifestyle, which is the significance of this finding.

We are very grateful for your assessment of our work. As you correctly note, irrespective of the tribe to which *Mesosymbion* belongs, the fossil has symbiont morphology, and it is this that provides the key evidence for social parasitism of Cretaceous social insect colonies. We favour the view that *Mesosymbion's* morphology is likely a specialized adaptation to life in colonies, rather than a preadaptation for it, since its suite of defensive traits are not seen in free-living aleocharines, but are observed frequently among the socially parasitic ones.

Reviewer #4:

This is an interesting study which give compelling evidence for social parasitism by rove beetles already about 100 Ma ago. The preservation of the specimen is remarkable and all important characters are clearly visible and very well documented. The conclusions are convincing and well-balanced. The references are appropriate, even recent paper on termite and ant evolution from 2016 are considered. I only have a few additional/critical comments:

On page 3 in the first paragraph you write: "However, undisputed evidence that Mesozoic termites and ants were definitively eusocial has been lacking^{21,28-30}, creating uncertainty as to whether either taxon formed colonies." I do not agree, because meanwhile we have clear evidences for such an early eusociality and indeed, in the next paragraph you write: "Recent studies of stem-group ants and termites in Burmese amber report clear evidence of advanced social organization in both insect groups by the mid-Cretaceous^{21,25}." Here you contradict yourself.

Thank you for your careful reading of our paper and apologies for the confusion here. We meant "until recently", clear evidence for eusociality had not been forthcoming — hence it was hard to know if Mesozoic aleocharines could have been social parasites. Now, the recent papers this year by Engel et al and Barden and Grimaldi appear to finally provide strong evidence for eusociality of Cretaceous ants and termites. We have modified the text to avoid appearing to contradict ourselves.

On page 9 in the last paragraph you use the term "ecological niche", but what you mean is that eusocial colonies present a new "resource" for exploitation by rove beetles.

We have changed the text accordingly.

REVIEWERS' COMMENTS:

Reviewer #2 (Remarks to the Author):

I am generally very happy with the revised manuscript and supplementary material.

Otherwise I only have very minor comments:

Page 6, last line: delete "and"

Page 7, line 11: add "the" before "extended"

Page 8, line 2: explain "explanate" for the non-specialist

Page 8, line 6: add "the" before "typical"

Page 9, line 11: add "the" or "the tribe" before "Deinopsini"

Page 9, lines 23-25: does this unpublished data form part of a manuscript in preparation?

Reviewer #3 (Remarks to the Author):

Nicely revised.